

# The effects of captive versus wild rearing environments on long bone articular surfaces in common chimpanzees (*Pan troglodytes*)

Kristi L. Lewton[1,2]

[1] Department of Integrative Anatomical Sciences, University of Southern California, Los Angeles, CA, United States of America

[2] Department of Biological Sciences, University of Southern California, Los Angeles, CA, United States of America

## ABSTRACT

The physical environments of captive and wild animals frequently differ in substrate types and compliance. As a result, there is an assumption that differences in rearing environments between captive and wild individuals produce differences in skeletal morphology. Here, this hypothesis is tested using a sample of 42 captive and wild common chimpanzees (*Pan troglodytes*). Articular surface areas of the humerus, radius, ulna, femur, and tibia were calculated from linear breadth measurements, adjusted for size differences using Mosimann shape variables, and compared across sex and environmental groups using two-way ANOVA. Results indicate that the articular surfaces of the wrist and knee differ between captive and wild chimpanzees; captive individuals have significantly larger distal ulna and tibial plateau articular surfaces. In both captive and wild chimpanzees, males have significantly larger femoral condyles and distal radius surfaces than females. Finally, there is an interaction effect between sex and rearing in the articular surfaces of the femoral condyles and distal radius in which captive males have significantly larger surface areas than all other sex-rearing groups. These data suggest that long bone articular surfaces may be sensitive to differences experienced by captive and wild individuals, such as differences in diet, body mass, positional behaviors, and presumed loading environments. Importantly, these results only find differences due to rearing environment in some long bone articular surfaces. Thus, future work on skeletal morphology could cautiously incorporate data from captive individuals, but should first investigate potential intraspecific differences between captive and wild individuals.

Corresponding author
Kristi L. Lewton, lewton@usc.edu

## INTRODUCTION

Morphological studies on primates tend to focus on osteological samples that derive from wild-caught individuals because of the assumption that the skeletons of individuals raised in captive environments differ from those living in wild or large range sanctuary environments (e.g., *Albrecht, 1982*). This perception is especially true among studies of great apes given the potentially large differences in wild versus captive ape environments. Chimpanzees in the

wild, for example, have large day ranges (e.g., 7.8–14.9 km$^2$, *Chapman & Wrangham, 1993*), use varied, three-dimensionally complex arboreal substrates as well as terrestrial substrates (*Doran & Hunt, 1994*), and males engage in large territory patrols (*Mitani & Watts, 2005*) and large-scale hunting (e.g., *Stanford et al., 1994*). In captive environments, enclosure size is limited and less three-dimensionally complex, and substrate types and compliance can be quite different than those found in the wild (i.e., less variation in heights, inclines, and compliance of simulated arboreal substrates, and increased hard surfaces such as rocks and concrete in captive settings). Because of this, and other issues related to potential differences in nutrition, growth, and physiology, skeletons of wild-caught individuals have been preferred for studies of morphological variation.

A handful of studies have investigated whether there are morphological differences between captive and wild osteological specimens (*McPhee, 2004*; *Zuccarelli, 2004*; *O'Regan & Kitchener, 2005*; *Bello-Hellegouarch et al., 2013*; *Hartstone-Rose et al., 2014*; *Antonelli et al., 2016*; *Kapoor et al., 2016*; *Turner et al., 2016*). Studies on cranial anatomy have shown differences in both size and shape of the skull in mice and felids (*McPhee, 2004*; *Zuccarelli, 2004*; *Hartstone-Rose et al., 2014*). The felid studies in particular have suggested that differences in skull shape between captive and wild individuals are related to differences in masticatory loading resulting from dietary differences. Studies on postcrania, however, have revealed that there are no differences between captive and wild specimens in the scapula of hominoids (*Bello-Hellegouarch et al., 2013*) nor in the lengths of long bones of vervet monkeys (*Turner et al., 2016*). The effects of captivity on postcranial joint surfaces, however have not previously been investigated. Given that long bone joint surfaces are responsible for load-bearing, and given that felids exhibit differences in skull shape related to masticatory loading differences, it is reasonable to hypothesize that differences in captive and wild loading environments may affect the articular surfaces of long bones.

Although diaphyseal cross-sectional geometry of long bones is phenotypically plastic and can respond to loading regimes (e.g., *Goodship, Lanyon & McFie, 1979*; *Burr et al., 1982*; *Ruff & Hayes, 1983*; *Schaffler et al., 1985*; *Ruff & Runestad, 1992*), the effects of altered loading regimes on long bone articular surfaces have been less well studied. Compared to diaphyseal dimensions, long bone epiphyses are more morphologically constrained and less labile (*Ruff, 1988*; *Ruff, Scott & Liu, 1991*; *Lieberman, Devlin & Pearson, 2001*). However, previous work has not compared long-term effects of altered loading environments on long bone articular surfaces.

The aim of this study is to test the hypothesis that long bone articular surfaces differ between captive and wild chimpanzees.

## MATERIALS AND METHODS

The sample includes 42 *Pan troglodytes* adult individuals (20 captive-raised and 22 wild, Table 1). Adulthood was assessed by long bone epiphyseal fusion. Only captive individuals aged 10 years or older that exhibited epiphyseal fusion were included in the study (an exception was made to include a 9-year-old individual that exhibited complete epiphyseal fusion). Individuals categorized as "captive" were captive-raised; most were born in captivity, but some were wild-born and brought to the United States during infancy.

**Table 1  Sample.**

| Wild, $N = 22$ | | | Captive, $N = 20$ | | | | |
|---|---|---|---|---|---|---|---|
| ID | Sex | Collection | ID | Sex | Environment | Age | Collection |
| HTB 1056 | M | CMNH | AP | F | Zoo | 30 | PFA |
| HTB 1708 | M | CMNH | CQ | F | PFA | 19 | PFA |
| HTB 1720 | F | CMNH | JE | F | PFA | 18 | PFA |
| HTB 1713 | F | CMNH | KI | F | PFA | 14 | PFA |
| HTB 1718 | M | CMNH | PR | F | PFA | 20 | PFA |
| HTB 1722 | M | CMNH | TR | F | PFA | 17 | PFA |
| HTB 1759 | F | CMNH | DO | F | PFA | 25 | PFA |
| HTB 1739 | M | CMNH | JU | F | Zoo | 22 | PFA |
| HTB 1723 | F | CMNH | LO | F | Pet | Ad. | PFA |
| HTB 1748 | F | CMNH | SA | F | Pet | Ad. | PFA |
| HTB 1758 | M | CMNH | BB | M | PFA | 16 | PFA |
| HTB 1770 | F | CMNH | PD | M | PFA | 18 | PFA |
| HTB 1775 | F | CMNH | LI | M | Other | 11 | PFA |
| HTB 1843 | F | CMNH | CC | M | Zoo | 27 | PFA |
| HTB 1853 | F | CMNH | CH | M | Zoo | Ad. | ASU |
| HTB 2071 | F | CMNH | PN | M | Zoo | 15 | ASU |
| HTB 2771 | F | CMNH | DA | M | Zoo | 21 | ASU |
| HTB 3552 | M | CMNH | TO | M | Zoo | 10 | ASU |
| HTB 2072 | M | CMNH | ED | M | Other | 17 | ASU |
| HTB 1882 | M | CMNH | SM | M | PFA | 9 | PFA |
| HTB 2026 | M | CMNH | | | | | |
| HTB 2746 | M | CMNH | | | | | |

**Notes.**
CMNH, Cleveland Museum of Natural History; PFA, Primate Foundation of Arizona; ASU, Arizona State University.

Fifteen (10 females, five males) of the captive-raised individuals were from the Primate Foundation of Arizona (PFA) skeletal collection, for which age at death, sex, cause of death, and life history information are known. Five additional adult male captive-raised chimpanzee skeletons were measured at the School of Human Evolution and Social Change at Arizona State University. The wild sample includes 22 adult (11 males, 11 females) skeletons from the Cleveland Museum of Natural History.

Measurements taken on the humerus, radius, ulna, femur, and tibia of each individual include linear dimensions (anteroposterior, mediolateral, and superoinferior) of the proximal and distal articular surfaces and total bone length. Each linear variable was measured three times and the average recorded for analysis. Articular surface areas were estimated using the linear dimensions following *Ruff (2002)* and *Ruff (2003)* published equations (Table 2). Bone lengths were measured in standard orientation following *Buikstra & Ubelaker (1994)*. Each surface area variable was size-standardized and converted to a Mosimann shape variable using a dimensionless ratio of the square root of the area variable to the geometric mean (*Mosimann, 1970*; *Jungers, Falsetti & Wall, 1995*). The geometric mean consisted of 15 variables: five bone lengths and the square roots of each articular

**Table 2  List of linear measures and equations used to calculate articular surface areas (from *Ruff, 2002*).**

| Measurement | Abbreviation/Equation |
|---|---|
| Humerus length | HL |
| Humeral head breadth | |
|     Anteroposterior | HHAP |
|     Superoinferior | HHSI |
| Humeral head depth | HHDP |
| Humeral trochlea breadth | |
|     Mediolateral | TRML |
|     Superoinferior | TRSI |
|     Anteroposterior | TRAP |
| Distal articular surface total breadth | HDML |
| Humeral capitulum breadth | |
|     Mediolateral | CPML |
|     Superoinferior | CPSI |
| Humeral capitulum depth | CPDP |
| **Humeral head surface area** | $\text{HHSA} = 3.14*((0.0625*(\text{HHSI} + \text{HHAP})^2) + \text{HHDP}^2)$ |
| Trochlea surface area | $\text{TRSA} = \text{TRAP}*\text{TRML}*\text{ACOS}(1 - ((2*\text{TRSI})/\text{TRAP}))$ |
| Capitulum surface area | $\text{CPSA} = \text{CPSI}*\text{CPML}*\text{ACOS}(1 - ((2*\text{CPDP}/\text{CPSI}))$ |
| **Total humeral distal surface area** | $\text{HDSA} = \text{TRSA} + \text{CPSA}$ |
| Radius length | RL |
| Radial head breadth | |
|     Anteroposterior | RHAP |
|     Mediolateral | RHML |
| Radiocarpal surface breadth | |
|     Anteroposterior | RCAP |
|     Mediolateral | RCML |
| **Radial head surface area** | $\text{RHSA} = 0.785*(\text{RHML}*\text{RHAP})$ |
| **Radiocarpal surface area** | $\text{RCSA} = \text{RCML}*\text{RCAP}$ |
| Ulna length | UL |
| Trochlear notch breadth | |
|     Mediolateral | UTML |
|     Superoinferior | UTSI |
| Trochlear notch depth | UTDP |
| Ulnar distal articular surface breadth | |
|     Anteroposterior | UCAP |
|     Mediolateral | UCML |
| **Trochlear notch surface area** | $\text{UTSA} = \text{UTSI}*\text{UTML}*\text{ACOS}(1 - ((2*\text{UTDP})/\text{UTSI}))$ |
| **Ulnar distal articular surface area** | $\text{UCSA} = 0.785*(\text{UCML}*\text{UCAP})$ |
| Femur length | FL |
| Femoral head breadth | |
|     Anteroposterior | FHAP |
|     Superoinferior | FHSI |

**Table 2** (*continued*)

| Measurement | Abbreviation/Equation |
|---|---|
| Femoral head depth | FHDP |
| Femoral condyle breadth | |
|    Medial condyle anteroposterior | MCAP |
|    Lateral condyle anteroposterior | LCAP |
|      Total mediolateral breadth | FCML |
|    Medial condyle superoinferior | MCSI |
|    Lateral condyle superoinferior | LCSI |
| Femoral condyle depth | |
|    Medial | MCDP |
|    Lateral | LCDP |
| **Femoral head surface area** | FHSA = 1.57*FHDP*(FHSI + FHAP) |
| **Femoral condyle surface area** | |
|    Medial | MCSA = MCSI*MCML*ACOS(1 − ((2*MCDP)/MCSI)) |
|    Lateral | LCSA = LCSI*LCML*ACOS(1 − ((2*LCDP)/LCSI)) |
|    Total | FCSA = MCSA + LCSA |
| Tibia length | TL |
| Tibial plateau breadth | |
|    Medial surface ML | MPML |
|    Lateral surface ML | LPML |
|      Total tibial plateau ML breadth | TPML |
|    Medial surface AP | MPAP |
|    Lateral surface AP | LPAP |
| Tibiotalar surface breadth | |
|    Anteroposterior | TTAP |
|    Mediolateral | TTML |
| **Tibial plateau surface area** | |
|    Medial | MPSA = 0.785*(MPML*MPAP) |
|    Lateral | LPSA = 0.785*(LPML*LPAP) |
|    Total | TPSA = MPSA + LPSA |
| **Tibiotalar surface area** | TTSA = TTML*TTAP |

**Notes.**
  Bold denotes the articular surface areas on which all analyses were performed.

surface area. Means and standard deviations for articular surface areas, bone lengths, and geometric means are listed in Table 3 (raw data are in Table S1).

Principal component analysis (PCA) was used to explore multivariate variation using shape variables (area$^{1/2}$/geometric mean). Factor loadings from the PCA were examined to determine which articular surfaces most influenced differences among sex-rearing groups. To test the hypothesis that captive and wild chimpanzees differ in articular surface areas, a multivariate analyses of variance (MANOVA) with sex, rearing, and sex*rearing effects was performed to yield a whole-model $F$-test for each of the three effects. This test determined that there were statistically significant sex, rearing, and sex*rearing effects within the sample (Table 4). This analysis was followed by univariate two-way analyses of variance (ANOVA) on each scaled articular surface area with rearing and sex as factors, each with

**Table 3 Means and standard deviations of absolute articular surface areas (mm²), the square root of articular surfaces scaled by geometric mean, bone lengths (mm), and geometric mean (mm) for each sex-rearing group.** Percentage differences between captive and wild sex-specific groups are also shown.

| | Captive | | | | Wild | | | | Percentage difference (captive–wild) | |
| | Female | | Male | | Female | | Male | | Female | Male |
| | Mean (mm²) | SD | Mean (mm²) | SD | Mean (mm²) | SD | Mean (mm²) | SD | | |
|---|---|---|---|---|---|---|---|---|---|---|
| Femoral head (FHSA) | 2223.1 | 352.6 | 2451.5 | 333.5 | 2408.8 | 467.1 | 2649.3 | 255.1 | −8.4% | −8.1% |
| FHSA$^{1/2}$/GM | 0.775 | 0.05 | 0.775 | 0.04 | 0.801 | 0.04 | 0.794 | 0.04 | −3.3% | −2.4% |
| Femoral condyles (FCSA) | 1991.0 | 259.5 | 2511.1 | 316.7 | 2106.9 | 288.3 | 2347.9 | 215.3 | −5.8% | 6.5% |
| FCSA$^{1/2}$/GM | 0.734 | 0.03 | 0.786 | 0.04 | 0.750 | 0.1 | 0.746 | 0.02 | −2.2% | 5.0% |
| Femur length | 285.4 | 24.8 | 289.2 | 15.8 | 292.1 | 15.7 | 302.9 | 14.5 | −2.4% | −4.8% |
| Tibial plateau (TPSA) | 1138.6 | 165.9 | 1218.0 | 99.4 | 1043.8 | 120.3 | 1149.1 | 166.4 | 8.3% | 5.7% |
| TPSA$^{1/2}$/GM | 0.553 | 0.02 | 0.549 | 0.01 | 0.527 | 0.02 | 0.522 | 0.03 | 4.7% | 5.0% |
| Distal tibia (TTSA) | 467.5 | 70.1 | 495.8 | 53.1 | 463.3 | 76.1 | 549.7 | 61.8 | 0.9% | −10.9% |
| TTSA$^{1/2}$/GM | 0.356 | 0.03 | 0.349 | 0.02 | 0.353 | 0.02 | 0.361 | 0.02 | 0.9% | −3.4% |
| Tibia length | 245.9 | 21.3 | 246.9 | 17.2 | 246.0 | 17.3 | 257.5 | 12.3 | −0.1% | −4.3% |
| Humeral head (HHSA) | 2062.8 | 268.3 | 1927.2 | 225.8 | 1990.0 | 252.5 | 2368.3 | 301.0 | 3.5% | −22.9% |
| HHSA$^{1/2}$/GM | 0.747 | 0.03 | 0.689 | 0.03 | 0.731 | 0.03 | 0.750 | 0.1 | 2.2% | −8.9% |
| Distal humerus (HDSA) | 1517.2 | 251.2 | 1713.0 | 133.4 | 1482.2 | 148.5 | 1734.6 | 200.0 | 2.3% | −1.3% |
| HDSA$^{1/2}$/GM | 0.637 | 0.03 | 0.649 | 0.03 | 0.629 | 0.02 | 0.642 | 0.03 | 1.2% | 1.1% |
| Humerus length | 291.1 | 24.7 | 288.1 | 18.6 | 297.5 | 18.0 | 309.0 | 13.7 | −2.2% | −7.3% |
| Radial head (RHSA) | 427.9 | 49.7 | 494.4 | 45.1 | 457.9 | 65.7 | 500.2 | 42.5 | −7.0% | −1.2% |
| RHSA$^{1/2}$/GM | 0.342 | 0.01 | 0.349 | 0.01 | 0.351 | 0.01 | 0.345 | 0.01 | −2.6% | 1.0% |
| Distal radius (RCSA) | 407.6[a] | 51.1 | 553.4 | 58.8 | 420.6 | 44.5 | 499.5 | 59.2 | −3.2% | 9.7% |
| RCSA$^{1/2}$/GM | 0.326 | 0.03 | 0.369 | 0.02 | 0.335 | 0.02 | 0.345 | 0.02 | −2.9% | 6.6% |
| Radius length | 268.5 | 18.5 | 271.0 | 17.8 | 269.5 | 14.9 | 287.0 | 14.2 | −0.4% | −5.9% |
| Proximal ulna (UTSA) | 952.9 | 146.8 | 1115.3 | 151.5 | 952.4 | 125.1 | 1088.3 | 132.2 | 0.1% | 2.4% |
| UTSA$^{1/2}$/GM | 0.506 | 0.02 | 0.523 | 0.03 | 0.505 | 0.04 | 0.509 | 0.03 | 0.1% | 2.7% |
| Distal ulna (UCSA) | 125.9 | 19.5 | 143.5 | 25.9 | 107.3 | 13.7 | 122.4 | 15.3 | 14.7% | 14.7% |
| UCSA$^{1/2}$/GM | 0.184 | 0.01 | 0.186 | 0.01 | 0.171 | 0.02 | 0.170 | 0.01 | 7.1% | 8.6% |
| Ulna length | 280.5 | 18.7 | 287.7 | 17.9 | 283.6 | 13.5 | 301.6 | 13.8 | −1.1% | −4.9% |
| Geometric mean (GM) | 60.7[a] | 3.6 | 63.692 | 2.77 | 61.0 | 2.8 | 64.8 | 2.2 | −0.4% | −1.7% |

**Notes.**
[a]Excludes DO due to pathology.

two levels. Sex was included as a main effect because chimpanzees are sexually dimorphic in body mass (*Smith & Jungers, 1997*, although they are not particularly dimorphic in skeletal size, *Gordon, Green & Richmond, 2008*). Thus, this statistical model allows the examination of the potential interaction effect between sex and rearing. When there was a significant sex*rearing interaction effect, *post hoc* pairwise comparisons were performed using Tukey's HSD tests. All analyses were performed in JMP Pro 13 (SAS Institute Inc., Cary, NC, USA). Although ANOVA is relatively robust to violations of assumptions of parametric statistics, the data were examined for violations of normality (using normal probability plots), homogeneity of variances (using Levene's test), and potential outliers

**Table 4** **Results of analyses of variance on scaled articular surfaces.** MANOVA (with sex, rearing, and sex-rearing interaction effects) results are followed by univariate two-way ANOVA results.

| | F | p |
|---|---|---|
| **Whole model MANOVA (Wilk's lambda)** | 2.83 | **<0.0001** |
| Sex | 2.95 | **0.01** |
| Rearing | 4.08 | **0.001** |
| Sex*Rearing | 2.59 | **0.02** |
| **FHSA$^{1/2}$/GM** | 0.97 | 0.42 |
| Sex | 0.07 | 0.79 |
| Rearing | 2.75 | 0.11 |
| Sex*Rearing | 0.07 | 0.79 |
| **FCSA$^{1/2}$/GM** | 5.83 | **0.002** |
| Sex | 7.11 | **0.01** |
| Rearing | 1.70 | 0.20 |
| Sex*Rearing | 9.41 | **0.004** |
| **TPSA$^{1/2}$/GM** | 5.75 | **0.002** |
| Sex | 0.53 | 0.47 |
| Rearing | 16.70 | **0.0002** |
| Sex*Rearing | 0.01 | 0.91 |
| **TTSA$^{1/2}$/GM** | 0.63 | 0.60 |
| Sex | 0.01 | 0.93 |
| Rearing | 0.46 | 0.50 |
| Sex*Rearing | 1.41 | 0.24 |
| **HHSA$^{1/2}$/GM** | 4.34 | **0.01** |
| Sex | 2.14 | 0.15 |
| Rearing | 2.85 | 0.10 |
| Sex*Rearing | 8.40 | **0.006** |
| **HDSA$^{1/2}$/GM** | 1.06 | 0.38 |
| Sex | 2.31 | 0.14 |
| Rearing | 0.86 | 0.36 |
| Sex*Rearing | 0.00 | 0.96 |
| **RHSA$^{1/2}$/GM** | 1.13 | 0.35 |
| Sex | 0.04 | 0.84 |
| Rearing | 0.53 | 0.47 |
| Sex*Rearing | 2.85 | 0.10 |
| **RCSA$^{1/2}$/GM** | 7.42 | **0.0005** |
| Sex | 13.95 | **0.0006** |
| Rearing | 4.03 | 0.05 |
| Sex*Rearing | 4.75 | **0.04** |
| **UTSA$^{1/2}$/GM** | 1.02 | 0.39 |
| Sex | 1.65 | 0.21 |
| Rearing | 0.81 | 0.37 |
| Sex*Rearing | 0.69 | 0.41 |

**Table 4** (*continued*)

| | *F* | *p* |
|---|---|---|
| **UCSA$^{1/2}$/GM** | 5.71 | **0.003** |
| Sex | 0.02 | 0.88 |
| Rearing | 16.94 | **0.0002** |
| Sex*Rearing | 0.17 | 0.68 |

**Notes.**
 *p*-values in bold are significant at alpha = 0.05.

(by examining studentized residuals), and were not found to violate these assumptions. One individual (DO) exhibited pathological morphology of the distal radius, and was excluded from the distal radius articular surface area analyses as well as the PCA.

## RESULTS

### Analyses of variance

The two-way ANOVAs revealed that five of the ten articular surface area shape variables exhibited significant differences in rearing, sex, or the interaction between sex and rearing (these are the humeral head, distal radius and ulna, femoral condyles, and tibial plateau; Table 4). There were significant differences due to rearing in the tibial plateau and distal ulna; captive individuals have relatively larger articular surfaces than wild individuals (Table 4, Figs. 1 & 2). Both the tibial plateau and the distal ulna only exhibited differences between captive and wild individuals without evidence of sex differences or an interaction effect between sex and rearing (Table 4). When scaled by size, the tibial plateau is 4.7% larger in captive females and 5% larger in captive males compared to their wild conspecifics. Similarly, the distal ulna is 7.1% larger in captive females and 8.6% larger in captive males compared to wild individuals (Table 3).

There were significant differences due to sex in the femoral condyles and distal radius; males have relatively larger femoral condyles and radiocarpal surfaces than females (Figs. 3 and 4). Lastly, there was a significant sex*rearing interaction effect in the femoral condyles, humeral head, and distal radius (Tables 4 and 5). For each of these interaction effects, captive males were significantly different from other sex-rearing groups (Table 5). Specifically, for the femoral condyles and distal radius, captive males have relatively larger articular surface shape variables than all other sex-rearing groups (Figs. 2 and 4). The scaled femoral condyles are 4.6% larger in captive males than in wild females, 5% larger than in wild males, and 6.6% larger than in captive females. The scaled distal radius is 6.6% larger in captive males than in wild males, 9.1% larger than wild females, and 9.4% larger than captive females. The only deviation from this general pattern of captive males having relatively larger articular surfaces is the humeral head, in which captive males have relatively smaller articular surfaces than wild males and captive females (Fig. 5).

### Principal component analysis

The first four principal components capture 76% of the variation within the sample (eigenvalues listed in Table 6). The plot of PC 1 on PC 2 captures 54.8% of sample variation and shows broad overlap between captive and wild specimens (Fig. 6). The only group that is differentiated in this plot is captive males, which separate from captive females
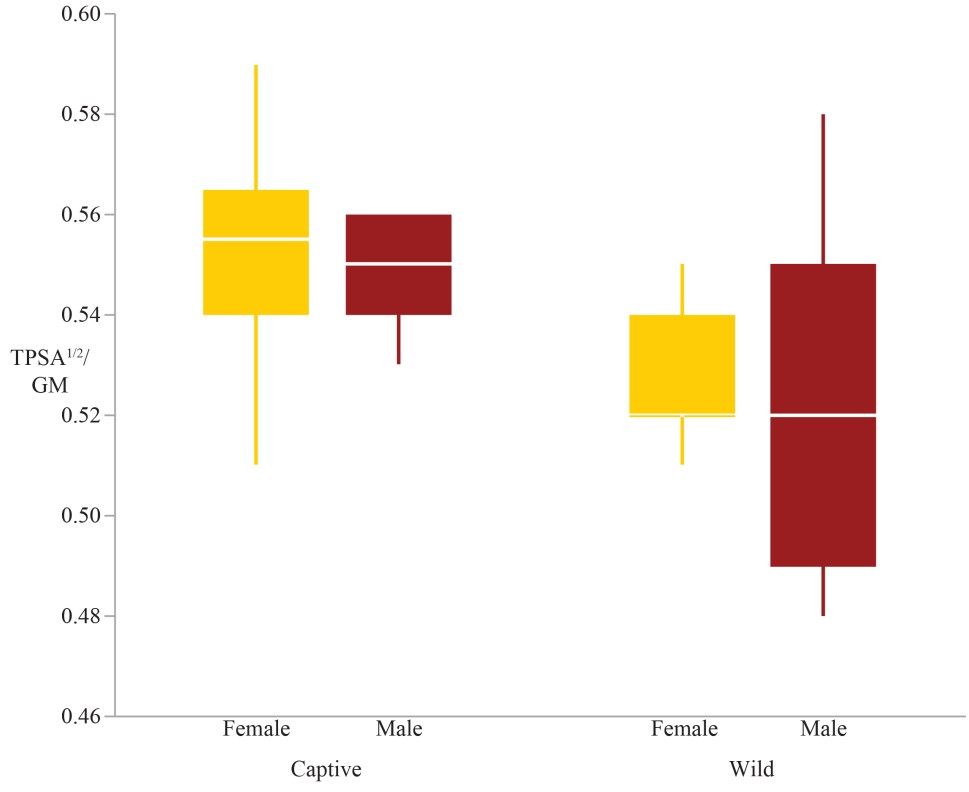

**Figure 1  Boxplot of the tibial plateau shape variable by sex and rearing.** Captive specimens have significantly greater tibial plateau scaled surfaces areas than wild specimens ($p = 0.0002$). There are no sex differences in tibial plateau area.

on PC 1, and show only minimal overlap with wild males and females. The three variables that load heavily on PC 1 are those that have a significant sex*rearing interaction effect (humeral head, distal radius, femoral condyles). The variable that loads heavily on PC 2 is femoral head surface area (Table 7). The plot of PC 3 on PC 4 captures an additional 21% of sample variation (Fig. 7). Neither PC 3 nor PC 4 differentiate groups by sex or rearing.

## DISCUSSION

This study demonstrates that there are some differences in scaled articular surface areas between captive and wild chimpanzees. Captive chimpanzees have relatively larger articular surfaces of the tibial plateau, distal radius, and distal ulna. The results also demonstrate an interaction effect between sex and rearing in which captive males have larger distal femur and radius articular surfaces than all other sex-rearing groups. Contrary to all other results, captive males have relatively smaller humeral heads than wild males. Although these data suggest that there are some differences in articular surface morphology between captive and wild chimpanzees, the differences are not ubiquitous, since only two out of ten variables differed solely due to rearing. This finding has two important implications: (1) long bone articular surfaces, like diaphyses, *can* differ due to different rearing environments, and (2) because only a few variables studied here differ between captive and wild chimpanzees,

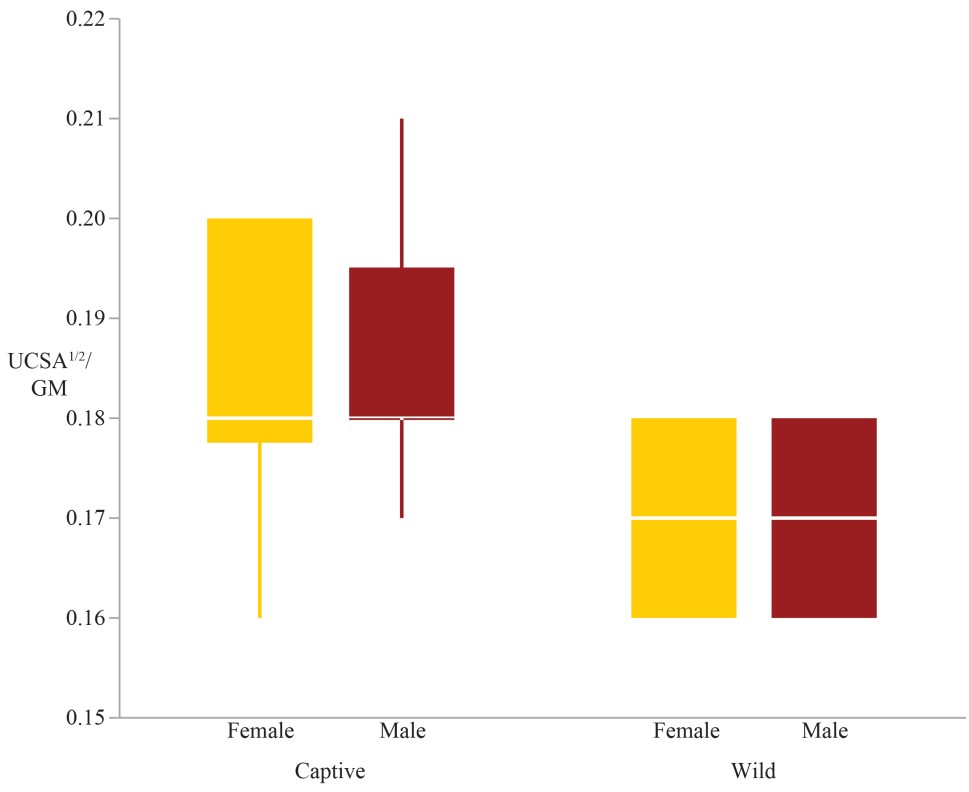

**Figure 2** **Boxplot of the distal ulna shape variable by sex and rearing.** Captive specimens have significantly greater distal ulna scaled surface areas than wild specimens ($p = 0.0002$). There are no sex differences in the distal ulna shape variable.

it may be appropriate to pool captive and wild specimens in analyses of bones that do not show statistically significant differences between the two groups. This would increase sample sizes, which is important when studying rare species such as great apes that are represented in skeletal collections by relatively few specimens.

## Behavioral, physiological, and environmental differences between captive and wild chimpanzee populations

Differences between captive and wild chimpanzees in the articular surfaces of the knee and wrist may be related to differences in positional behavior, differences in substrate types and compliance, or to a combination of these factors between these two groups. *Schwandt (2002)* found slight differences in the ontogeny of positional behavior in the PFA chimpanzees compared to wild chimpanzees from the Taï Forest, Côte d'Ivoire (*Doran, 1989*; *Doran, 1992*). The captive PFA chimpanzees knuckle-walked more often during youth than wild chimpanzees, used more suspensory behaviors during 6–9 years of age, and retained more variety in positional behavior repertoires compared to Taï Forest chimpanzees, whose positional behavior repertoires are static at six years of age (*Schwandt, 2002*). Given the role of the wrist in achieving knuckle-walking and suspensory behaviors, differences in frequencies of these behaviors, and concomitant effects on loading and/or joint range of motion, may affect the articular surfaces of the distal radius and ulna, as found here.

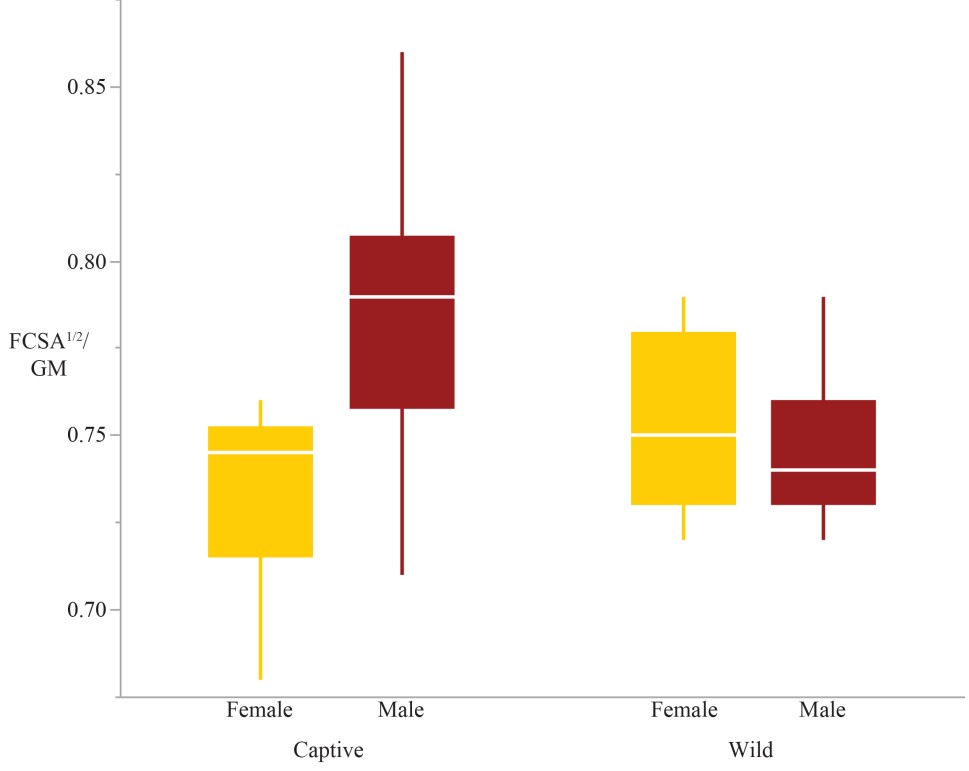

**Figure 3  Boxplot of the femoral condyles shape variable by sex and rearing.** There are significant sex and sex\*rearing effects; males have greater femoral condyle scaled surfaces than females ($p = 0.01$) and captive males have greater femoral condyle areas than all other groups ($p = 0.004$).

There are also environmental differences between captive and wild chimpanzees that may affect bony morphology. Most of the captive chimpanzee population was housed in enclosures with concrete flooring (as well as simulated arboreal substrates, *Schwandt, 2002*), which could affect joint surfaces. Previous research has demonstrated that long-term locomotion on concrete substrates affects the articular cartilage and joint congruency of the knee in sheep housed on tarmac compared to sheep housed in pasture (*Radin et al., 1982*). Perhaps the differences in tibial plateau articular surface area found here could be explained by altered loading regimes resulting from concrete substrates in the captive chimpanzee sample. Concrete surfaces might result in larger joint reaction forces than those that result from the more compliant arboreal and terrestrial substrates found in wild environments. However, some of the captive individuals studied here were not housed for the entirety of their lives at PFA, and the details of the substrates of their rearing environments are uncertain.

There may be other differences between captive and wild chimpanzees besides positional behaviors and physical environments that could affect long bone articular surfaces. In baboons, for example, it has been suggested that dietary differences between captive and wild individuals could be responsible for observed differences in somatic maturation rates, where captive animals may be "overfed," resulting in more energy that can be allocated to growth than in their wild counterparts (*Phillips-Conroy & Jolly, 1988*). Similar scenarios

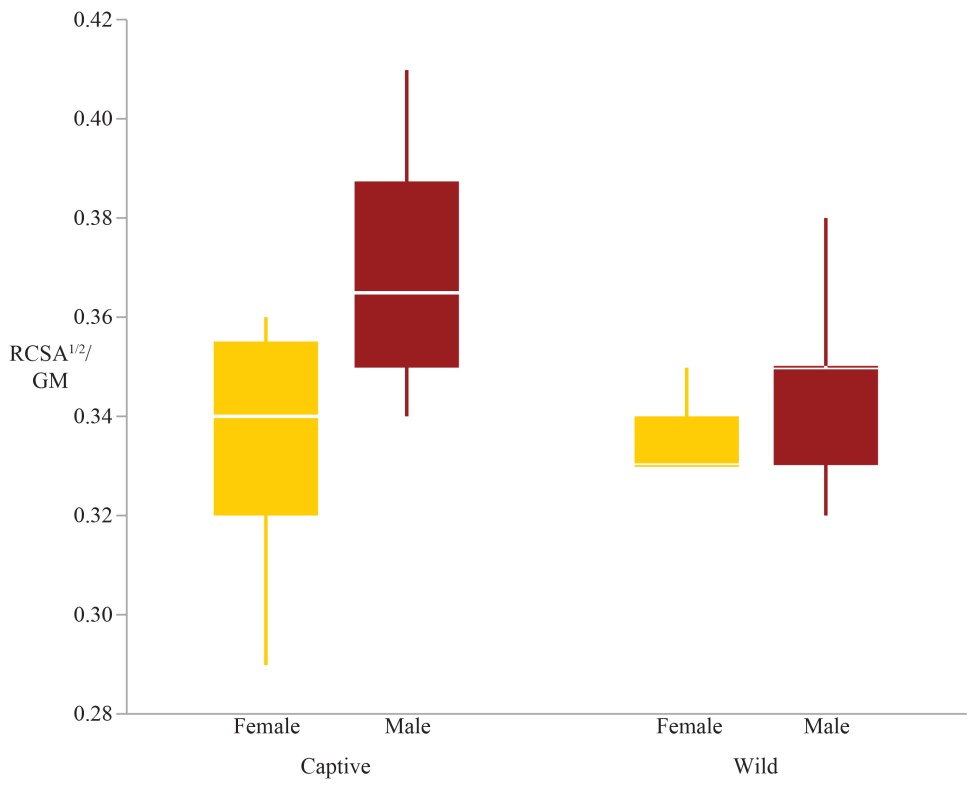

**Figure 4  Boxplot of the distal radius shape variable by sex and rearing.** This plot shows significant sex and sex*rearing interaction effects. Males have relatively greater distal radius scaled surface areas than females ($p = 0.0006$), and captive males have relatively greater areas than all other groups ($p = 0.04$).

have been suggested for chimpanzees. Wild chimpanzees in Gombe National Park spend just over 50% of their time foraging and feeding (*Wrangham, 1977*). Their diets consist mostly of fruit, but they eat a variety of other items including other aspects of plant material (nuts, seeds, leaves, flower blossoms, and piths) as well as honey, insects, and meat (*Boesch & Boesch-Achermann, 2000*). Captive chimpanzees, on the other hand, tend to eat a less varied diet that is higher in caloric content, and they tend to spend less time foraging and feeding than wild chimpanzees (*Pruetz & McGrew, 2001*). Perhaps the combined effects of reduced activity levels and higher-calorie diets in captive chimpanzees could result in the observed accelerated skeletal growth rates in which captive chimpanzees reach skeletal maturity from six months to up to three years before wild chimpanzees (*Zihlman, Bolter & Boesch, 2007*). However, if nutritional differences were responsible for the articular surface area differences found in this study, then one might expect to observe uniform differences between captive and wild specimens across all measured long bone epiphyses. That this study only found rearing differences in two out of ten measured articular surfaces (proximal tibia and distal ulna) suggests that the likely cause of skeletal differences is multifactorial and more complex than diet alone.

The differences in skeletal maturation rates between captive and wild chimpanzees demonstrated by Zihlman and colleagues (*2007*) suggests that exploring the effects of

**Table 5  Results of Tukey's HSD *post hoc* pairwise comparisons for significant sex-rearing interaction effects.**

| $FCSA^{1/2}/GM$ | CF | CM | WF | WM |
|---|---|---|---|---|
| CF | | | | |
| CM | **0.002** | | | |
| WF | 0.60 | **0.04** | | |
| WM | 0.77 | **0.02** | 0.99 | |
| $HHSA^{1/2}/GM$ | CF | CM | WF | WM |
| CF | | | | |
| CM | **0.02** | | | |
| WF | 0.83 | 0.13 | | |
| WM | 1.00 | **0.01** | 0.73 | |
| $RCSA^{1/2}/GM$ | CF | CM | WF | WM |
| CF | | | | |
| CM | **0.001** | | | |
| WF | 1.00 | **0.001** | | |
| WM | 0.63 | **0.02** | 0.67 | |

**Notes.**
*p*-values in bold are significant at alpha = 0.05.
CF, captive female; CM, captive male; WF, wild female; WM, wild male.

**Table 6  Eigenvalues of principal components.**

| PC | Eigenvalue | Cumulative % |
|---|---|---|
| 1 | 0.0027 | 32.1 |
| 2 | 0.0019 | 54.8 |
| 3 | 0.0011 | 67.4 |
| 4 | 0.0007 | 75.8 |
| 5 | 0.0006 | 83.3 |
| 6 | 0.0005 | 89.2 |
| 7 | 0.0004 | 93.6 |
| 8 | 0.0003 | 97.1 |
| 9 | 0.0002 | 99.0 |
| 10 | 0.0001 | 100 |

allometry on differences in skeletal size between captive and wild chimpanzees could provide additional explanatory power. Although a full allometric analysis is beyond the scope of this paper, a test of homogeneity of slopes of the regressions of the square root of each articular surface area on the geometric mean between the captive and wild samples demonstrates that most variables do not differ in their allometric coefficients of long bone articular surface areas (Table 8). The exception is the distal tibia, for which the slope test was significant. However, this variable does not differ by sex or rearing, so any allometric differences are likely not meaningful. Thus, there are likely additional hormonal, nutritional, size, or gestational differences that could affect the epiphyses of long bones. Future work investigating the likely multifactorial causes of increased somatic growth rates in captive chimpanzees would be valuable for understanding the biology of the genus as a whole.

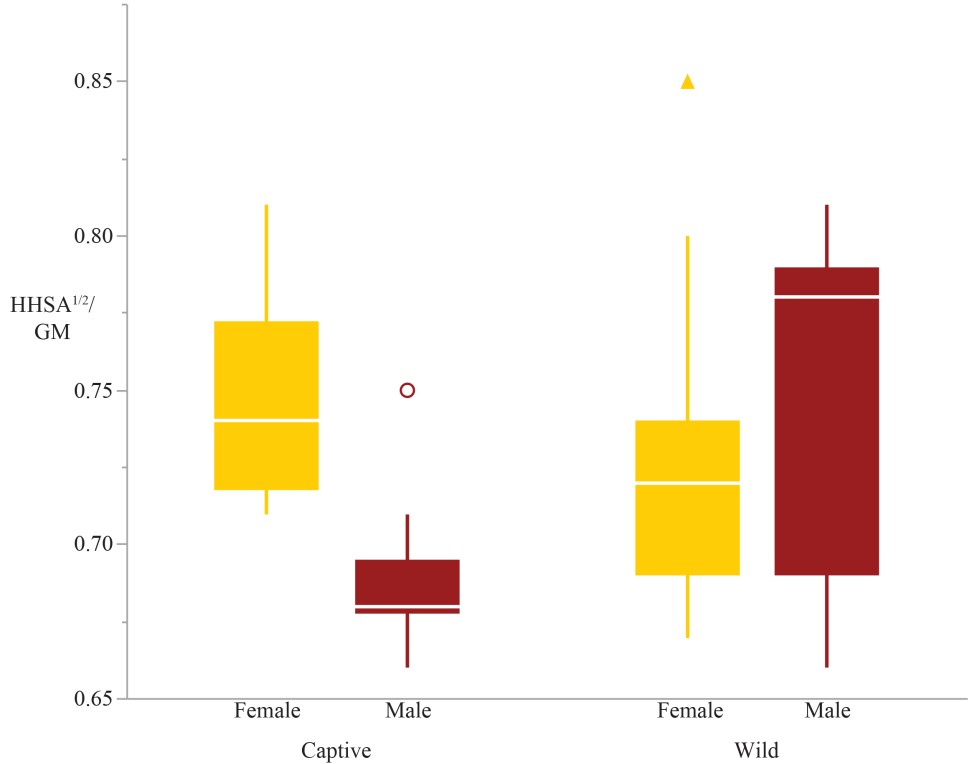

**Figure 5** **Boxplot of the humeral head shape variable by sex and rearing.** There is a significant sex*rearing interaction effect; captive males have smaller humeral head scaled surface areas than wild males ($p = 0.01$) and captive females ($p = 0.02$).

**Table 7** **Variable loading matrix for the ten principal components.**

|  | PC 1 | PC 2 | PC 3 | PC 4 | PC 5 | PC 6 | PC 7 | PC 8 | PC 9 | PC 10 |
|---|---|---|---|---|---|---|---|---|---|---|
| HHSA$^{1/2}$/GM | **0.961** | −0.029 | 0.253 | −0.004 | 0.033 | 0.078 | 0.034 | 0.058 | 0.002 | −0.012 |
| FHSA$^{1/2}$/GM | 0.055 | **0.970** | −0.141 | 0.112 | 0.115 | 0.099 | 0.032 | 0.014 | 0.001 | −0.013 |
| FCSA$^{1/2}$/GM | **−0.411** | 0.380 | **0.785** | −0.155 | −0.147 | −0.100 | −0.119 | −0.021 | −0.030 | −0.018 |
| HDSA$^{1/2}$/GM | −0.134 | −0.238 | 0.317 | **0.647** | **0.593** | −0.169 | −0.054 | −0.136 | −0.066 | 0.002 |
| UTSA$^{1/2}$/GM | −0.360 | −0.085 | 0.148 | **−0.587** | **0.636** | 0.106 | 0.266 | 0.079 | 0.065 | 0.012 |
| TPSA$^{1/2}$/GM | −0.394 | −0.241 | 0.247 | 0.203 | −0.138 | **0.756** | 0.143 | −0.264 | 0.048 | −0.029 |
| TTSA$^{1/2}$/GM | 0.291 | 0.204 | 0.050 | −0.036 | −0.232 | −0.404 | **0.657** | **−0.466** | −0.053 | 0.043 |
| RCSA$^{1/2}$/GM | **−0.502** | −0.111 | 0.214 | 0.433 | −0.202 | −0.063 | **0.448** | **0.493** | 0.091 | −0.079 |
| RHSA$^{1/2}$/GM | 0.066 | 0.223 | 0.304 | 0.277 | −0.077 | −0.048 | −0.082 | 0.021 | **0.668** | **0.561** |
| UCSA$^{1/2}$/GM | −0.221 | 0.008 | 0.087 | 0.027 | −0.079 | 0.331 | 0.178 | 0.327 | −0.714 | **0.422** |

**Notes.**
Bold indicates variables that load heavily (absolute value greater than 0.4) on each principal component.

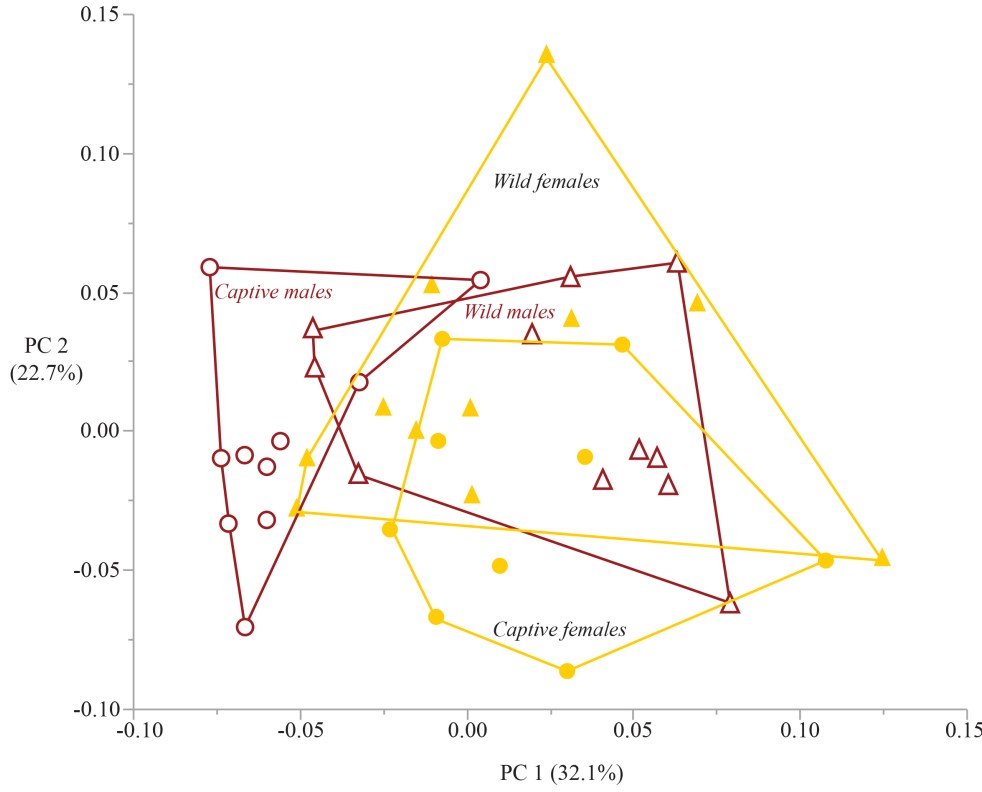

**Figure 6  Bivariate plot of principal components 1 and 2.** The plot of PC 1 on PC 2 demonstrates overlap among captive and wild individuals. PC 1 differentiates captive males from the other sex-rearing groups. PC 2 does not differentiate among the four sex-rearing groups. Circles are captive specimens, triangles are wild specimens. Filled symbols are females, open symbols are males. Convex hull polygons enclose each group.

**Table 8  Tests of homogeneity of slopes between captive and wild samples for regressions of the square root of each articular surface area on the geometric mean demonstrate no statistical differences.**

| | Captive | | | Wild | | | | | |
|---|---|---|---|---|---|---|---|---|---|
| **Square root of:** | **N** | **Slope** | **SE** | **N** | **Slope** | **SE** | **t** | **df** | **p** |
| Femoral head (FHSA) | 20 | 0.72 | 0.19 | 22 | 0.90 | 0.18 | 0.68 | 38 | 0.50 |
| Femoral condyles (FCSA) | 20 | 0.96 | 0.18 | 22 | 0.81 | 0.11 | 0.75 | 38 | 0.46 |
| Tibial plateau (TPSA) | 20 | 0.51 | 0.07 | 22 | 0.57 | 0.10 | 0.44 | 38 | 0.66 |
| Distal tibia (TTSA) | 20 | 0.21 | 0.09 | 22 | 0.46 | 0.08 | 2.11 | 38 | **0.04** |
| Humeral head (HHSA) | 20 | 0.45 | 0.16 | 22 | 0.53 | 0.22 | 0.28 | 38 | 0.78 |
| Distal humerus (HDSA) | 20 | 0.69 | 0.11 | 22 | 0.71 | 0.11 | 0.08 | 38 | 0.94 |
| Radial head (RHSA) | 20 | 0.36 | 0.04 | 22 | 0.32 | 0.06 | 0.58 | 38 | 0.56 |
| Distal radius (RCSA) | 19 | 0.31 | 0.13 | 22 | 0.41 | 0.06 | 0.75 | 37 | 0.46 |
| Proximal ulna (UTSA) | 20 | 0.65 | 0.10 | 22 | 0.48 | 0.12 | 1.09 | 38 | 0.28 |
| Distal ulna (UCSA) | 20 | 0.18 | 0.06 | 22 | 0.17 | 0.04 | 0.14 | 38 | 0.89 |

**Notes.**
$p$-value in bold is significant at alpha $= 0.05$.

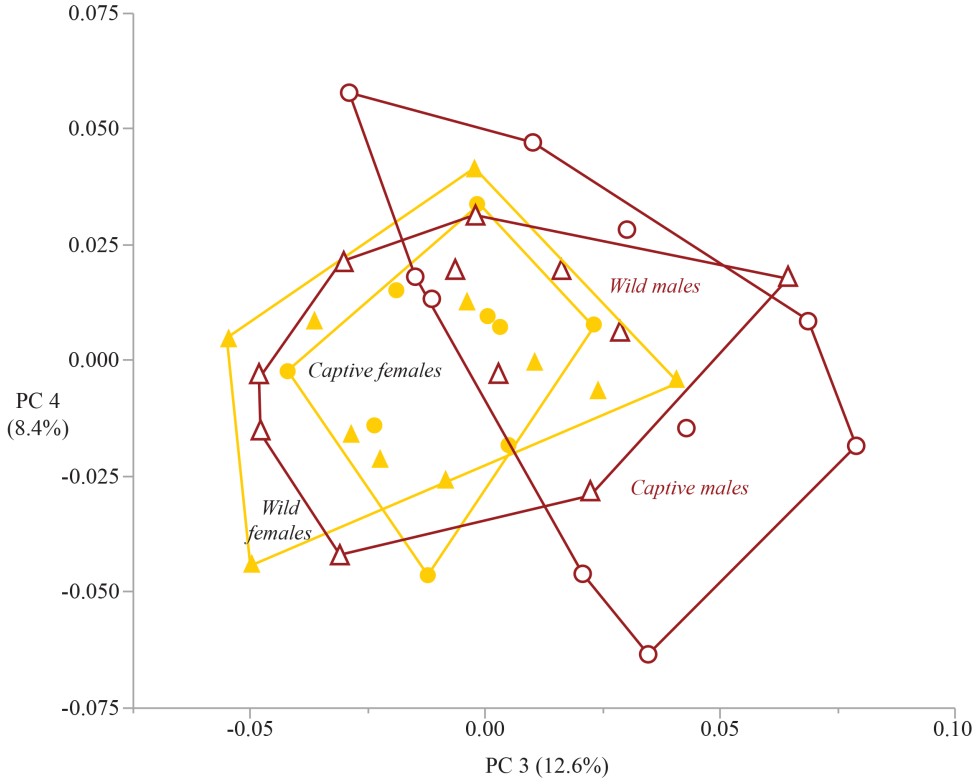

**Figure 7** **Bivariate plot of principal components 3 and 4.** The plot of PC 3 on PC 4 does not differentiate the sample by either sex or rearing. Circles are captive specimens, triangles are wild specimens. Filled symbols are females, open symbols are males. Convex hull polygons enclose each group.

## Environmental effects on long bone articular surfaces

In addition to providing more information about the potential differences between captive and wild osteological specimens, this study provides some broader information about how articular surfaces may respond to loading regimes. Some previous work has found that cartilage and subchondral bone thickness and hardness respond to loading (e.g., *Bouvier & Zimny, 1987*; *Murray et al., 2001*; *Plochocki, Riscigno & Garcia, 2006*, but see *Hammond et al., 2010*), and dimensions of joint surfaces can respond to loading by increasing in overall size (*Bouvier & Zimny, 1987*; *Plochocki, Riscigno & Garcia, 2006*, but see *Lieberman, Devlin & Pearson, 2001*). This study finds that captive chimpanzees exhibit relatively larger articular surface areas of parts of the knee and wrist joints compared to wild chimpanzees, which could potentially result from long-term housing on stiff, relatively noncompliant substrates. If larger articular surfaces in captive animals are related to different loading regimes, then one might expect that distal postcranial joints would be more susceptible to increases in peak substrate reaction forces. The wrist data from this study support this hypothesis, but the ankle data do not. Additionally, it is unclear why the knee joint would show articular changes while the elbow joint does not. Kinetic data on the effects of compliant versus noncompliant substrates on postcranial joint reaction forces are necessary to address these complex questions. Like other primates (e.g., *Kimura, Okada & Ishida,*

*1979*), chimpanzees experience larger hindlimb peak vertical forces during quadrupedalism than forelimb forces (*Reynolds, 1985*; *Demes et al., 1994*). Given this phenomenon, one might also expect that hindlimb joints would respond more readily to increasing loading regimes in captive animals.

## CONCLUSIONS

Skeletal collections of primates are rare and encompass endangered species for which data are dwindling (e.g., *Gordon, Marcus & Wood, 2013*). Therefore, using captive-reared specimens that are curated within existing skeletal collections, as well as imaging data (e.g., computed tomographic scans) of sanctuary or zoological specimens, may be important ways to increase sample sizes. Previous work on the effects of captivity on long bones did not find differences between captive and wild specimens in long bone lengths (*Turner et al., 2016*) or scapular shape (*Bello-Hellegouarch et al., 2013*), and the present study only found differences in some long bone articular surfaces (i.e., aspects of the knee and wrist). Taken together, this body of work suggests that there is no *a priori* reason to exclude captive individuals from morphological analyses. However, if one chooses to combine captive and wild specimens in a study of morphological variation, it would be prudent to first examine the data to determine whether significant differences exist between captive and wild specimens, or whether captive specimens are statistical outliers.

## ACKNOWLEDGEMENTS

Thank you to Lyman Jellema (Cleveland Museum of Natural History), and Jo Fritz and Elaine Videan (Primate Foundation of Arizona) for providing access to skeletal materials. This project was substantially improved by discussions with Kaye Reed, Mary Marzke, Dennis Young, Stephanie Meredith, and William Jungers. This protocol has been reviewed and approved by the Primate Foundation of Arizona, Institutional Animal Care and Use Committee (IACUC).

### Funding
The author received no funding for this work.

### Competing Interests
The author declares that there are no competing interests.

### Author Contributions
- Kristi L. Lewton conceived and designed the experiments, performed the experiments, analyzed the data, contributed reagents/materials/analysis tools, wrote the paper, prepared figures and/or tables, reviewed drafts of the paper.

### Data Availability
The raw data has been supplied in Table S1.

## Supplemental Information

Supplemental information for this article can be found online at http://dx.doi.org/10.7717/peerj.3668#supplemental-information.

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
