# Peer review of "The effects of captive versus wild rearing environments on long bone articular surfaces in common chimpanzees (Pan troglodytes)"

_PeerJ, doi:10.7717/peerj.3668_

## Round 0.1 · original submission · Major Revisions

You have received two very detailed reviews that will help you to greatly improve your manuscript. I strongly recommend you to explain your methods in greater detail and to review your statistical approach. Please, consider more deeply the possible effect of factors such as diet on growth patterns. Consider also to add the literature and modify your figure 6 as suggested by our reviewer.

·

Basic reporting

The manuscript shows a quite strait-forward analysis. Five linear variables are measured and rescaled according to a given equation. The text is clear and unambiguous. English language is correct. The introduction is short and shows the required context, since references on wild-captive skeletal comparisons are scarce. Literature is appropriate. Raw data is properly supplied in supplementary xlsx table. The results are supported by the analyses. However, this is a limited research that might be appropriate within a full research that merges captive/wild animals.

Experimental design

The manuscript deals the question of anatomical variation according to captivity/wild factor. Since cranial and humeral head analysis are available, articulation surfaces are analysed. The experimental design involves measuring distances and scaling the measurements for size differences given sexual dimorphism. A simple research question is defined: check for differences in articular surface areas between rearing groups. The methods should though be explained in greater detail:
The equations for rescaling measurements should be included (along with the reference already given Ruff C. 1988, for greater detail for readers and to allow checking the results in Table 2). In fact, there are clear mistakes in Table 2. Greater precision in decimal places should be provided for the Geometric Mean calibration. For instance, for FHSA1/2/GM, 0.8 captive females, and 0.8 wild females cannot result in a -3% difference (but 0% difference). There are many of these mistakes in this table that, in part providing 4 decimal places would deal with. In line 98 specify the equations for computing surface estimations. Are this equations equally applicable to captive and wild animals? Are captive animals larger because of better diets and thus have allometric larger articulation surfaces rather than due to activity patterns?
Regarding the statistical analysis, it would be more appropriate to male a multifactorial MANOVA (5 variables, 2 factors) instead of several one-way ANOVAs with a post-hoc for a combined SEX-REARING factor (as seems to be the case in Table 3). In SPSS this can be done with a MULTIVARIATE MULTIFACTOR MODEL (no post-hoc needed). This would be a more neat statistical approach. Some references regarding growth rates in captivity should be cited. Growth rate un wild gorillas have been published by Galbany J et al.
Indicate why young specimens (9, 10, 11 years old for captive) were included in the analyses. Could this affect the results? Were they fully grown-up as adults? Perhaps consider excluding them or at least indicate why were the included and indicate their positions in Figure 6. Do these speciemns change the dispersion of the box plots? Are they outliers?
Include the P values also for the n.s. (not significant) tests in Table 3. They are also informative. I suggest doing a Multifactorial MANOVA instead of these in Table 3.
In Figure 6 use convex-hulls or equiprobable ellipses for showing the dispersion range of each group. Otherwise it is difficult to follow the differences indicated in the caption.

Validity of the findings

The results obtained are clear. Some significant differences could be attributed to captivity, regardless of size and sexual dimorphism. However, these differences cannot clearly be attributed to any factor related to behavioral factors in captivity, rather than citing hardness of floor. You may elaborate more on the possible effect of factors such as diet on growth patterns
From line 160 to 168 results are repeated. Modify to go strait to the discussion.
Lines 169-173. In fact the results obtained suggest the opposite as in implication 2). You suggest here that captive and wild samples can be merged, but results indicate that if you are studying the wrist and the knee this may not be the case. This is well explained later, but inference 2 is giving a wrong impression.
Interpretations are made in terms of surface areas (rather than on linear measurements), but are these estimated variables accurate? Why not talk only about widths? Area estimations assume equal articular surface shapes, and this might no be the case. Why did you not measure both diameters of the surfaces? At present, this withdrawals are approached using 3D surface area measures (3D scanning and exact area measuring).
Line 207-208. Are these differences caused by differences in size rather than to posture regimes?
In fact that indicated in lines 218-221 cannot be tested, although they may be indeed be affecting the results.
Despite these suggestions, the research presented is fairly well designed and adds relevant information to this topic. The conclusions cannot be considered definitive and testing this hypothesis needs to be done on a one-by-one research trying to merge captive and wild samples to enlarge the sample sizes.

Additional comments

In this regard, a more appropriate approach for this paper would have been to focus on a specific research topic and the test if for the variables considered captive/wild samples can be merged and the proceed with the analysis. A see this paper as a methodological question within a more ambitious research design. However, this does not demerit the interest of the research made.

·

Basic reporting

One of the best written papers that I have reviewed on an initial review.

All this:

Clear and unambiguous, professional English used throughout.
Literature references, sufficient field background/context provided.
Professional article structure, figs, tables. Raw data shared.
Self-contained with relevant results to hypotheses.

Experimental design

Excellent design. Although the paper suffers - like ALL of our studies on the morphology of endangered species - from small overall statistical sample sizes, the sample is clearly adequate for findings of significance and is a reasonable representation of readily available individuals.

The question is important, well defined, and well explored.

Validity of the findings

The findings are mixed - ah biology! - and this diversity is well discussed. I do not question the findings at all, and the statistics have been conducted nearly exactly as I would have done them myself.

Additional comments

Review of: The effects of captive versus wild rearing environments on long bone articular surfaces in common chimpanzees (Pan troglodytes). Lewton. Summer 2017 submission for PeerJ.

This paper is excellent and should be published essentially as is. It is highly edited and much more refined than almost any paper that I have reviewed, and, more importantly, is about an important subject that has long required further study. I have literally no line edits (kudos to whomever you had edit this!), and have only two minor suggestions:

1) My students have published papers linking oral health measures to changes in morphological shape in captive and wild carnivores. Given how few studies there have been in actually quantifying the difference between captive and wild animals, it might be worth including those references as we build the body of literature. (I really look forward to citing your contribution – especially valuable given the primate focus!)

Antonelli TS, Leischner CL, Ososky JJ, and Hartstone-Rose A. 2016. The effect of captivity on the oral health of the critically endangered black-footed ferret (Mustela nigripes). Canadian Journal of Zoology 94(1):15-22.

Kapoor V, Antonelli T, Parkinson JA, and Hartstone-Rose A. 2016. Oral Health Correlates of Captivity. Research in Veterinary Science 107:213-219.


2) I am very visual in my thinking about this type of thing (aren’t many morphologists?), and therefore find Figure 6 the most compelling part of your paper. I would suggest two additions to this figure: a) I think the figure would really benefit from drawing the bounding lines (minimum convex units) around your four states and label them directly on the graph and not just in the caption. I also use JMP and think there is no way to do this in that program, but it is easy enough to add this to the figure in PPT or PSD. Also, b) although you talk about the loadings in the results, I think the discussion would benefit greatly from the addition of the actual Eigen values – best included as a table. Although not related to this figure specifically, I am curious to see the figures (and loadings) for the third and fourth PCs. I take you at your word that they do not separate the groupings, but I am curious nonetheless.

Lastly, a few notes to consider in your discussion: the humerus data is indeed very interesting. Is there any evidence that captive chimps (especially males) brachiate less? That might explain this finding. Also, as you allude to toward the end of the paper, I suspect that some of these morphological differences are due to life history effects – e.g., maturation rates differing between captive and wild animals. It would be interesting to see whether the effected joints fuse at different rates in captive and wild animals – maybe, for instance, the humeral head fuses quicker and the other joints fuse slower and the complex size differences are due to an allometric pattern based on simple linear growth rates. I don’t know if there is any data out there to support something like this (you know much more than I do about this anatomy), but it is interesting to think about. Lastly, the only thing that I struggled with in this paper is the issues surrounding size; I think I too would have used geometric mean exactly like you did to remove the influence of size, but this is still an oversimplification of morphology that is clearly affected by allometry. And when you have such a wide range of ages (or no known ages in the wilds – another complexity) along with relatively small sample sizes (though these are large for what they are!), having one or two super robust or gracile individuals could throw things off. However, I think you handled this as well as you could, and not only do I think that I would have handled these issues almost exactly as you did, I think the data are valuable enough that these confounding issues need to be excused. Though you could talk about them a bit more, if you are so inclined.

Great work! Really valuable contribution.

Sincerely – Adam Hartstone-Rose

---

## Round 0.2 · accepted · Accept

Many thanks for your careful consideration of all the suggestions of our reviewers.

·

Basic reporting

All my concerns and suggestions in the first review were fully considered and sufficiently respond to my indications, including statistical approach and discussion.

Experimental design

The experimental design is fully adequate and the modified disccusion is now fully adequate.

Validity of the findings

Data is robust and fully sustained with the new results provided in the reviwed version.

Additional comments

Great jab and review.